# A cross-sectional study of university students' mental health and lifestyle practices amidst the COVID-19 pandemic

Reem Hoteit [1]*, Imad Bou-Hamad[2], Sahar Hijazi[3], Dinah Ayna[4], Maya Romani[5], Christo El Morr[6]

1 Clinical Research Institute, Faculty of Medicine, American University of Beirut, Beirut, Lebanon, 2 Department of Business Information and Decision Systems, Suliman S. Olayan School of Business, American University of Beirut, Beirut, Lebanon, 3 Faculty of Social Sciences, Lebanese University, Saida, Lebanon, 4 Department of Psychiatry, Faculty of Medicine, American University of Beirut Medical Center, Beirut, Lebanon, 5 Department of Family Medicine, Faculty of Medicine, American University of Beirut Medical Center, Beirut, Lebanon, 6 School of Health Policy and Management, York University, Toronto, Canada

* rah84@aub.edu.lb

**Data Availability Statement:** We made the data publicly available by depositing it in a data bank maintained by The American University of Beirut's

## Abstract

### Objectives

University students are regarded as the backbone of society, and their mental health during a pandemic may have a substantial impact on their performance and life outcomes. The purpose of this study was to assess university students' mental health, specifically depression, anxiety, and stress, during Lebanon's extended COVID-19 pandemic, as well as the socio-demographic factors and lifestyle practices associated with it.

### Methods

An online anonymous survey assessed the rates of mental health problems during COVID-19, controlling for socio-demographics and other lifestyle practices, in 329 undergraduate and graduate university students. Instruments utilized were the Patient Health Questionnaire (PHQ-9) for depression, the Beck Anxiety Inventory (21-BAI) for anxiety, and the Perceived Stress Scale (PSS-10) for stress. The study employed descriptive statistics and multiple logistic regression models to analyze the association between depression, anxiety, and stress with sociodemographic and lifestyle factors. Results were evaluated using adjusted odds ratios and confidence intervals, with a significance level of 0.05.

### Results

Moderate to severe rates of depression, anxiety and stress among students were reported by 75.9%, 72.2%, and 89.3%, respectively. The odds of anxiety and stress were higher among women compared to men. Students who used private counseling services had higher odds of anxiety and stress than those who did not. Overall rated health was a major predictor of depression and anxiety, with the "poor" and "fair" overall-reported health groups having higher odds than the "Excellent" group. When compared to those who did not

libraries. To access the data, use the following:
DOI: http://hdl.handle.net/10938/24094.

**Funding:** The author(s) received no specific
funding for this work.

**Competing interests:** The authors have declared
that no competing interests exist.

smoke, students who increased their smoking intake had higher odds of depression, anxiety
and stress. Students who reduced their alcohol consumption had lower odds of anxiety com-
pared to those who did not consume alcohol. Students who reduced their physical activity
had higher odds than those who increased it. Finally, students who slept fewer than seven
hours daily had higher odds of depression than those who slept seven to nine hours.

## Conclusion

Our findings indicate a national student mental health crisis, with exceptionally high rates of
moderate to severe depression, anxiety, and stress. Factors such as gender, university pro-
gram, overall rated health, importance of religion in daily decisions, private counseling,
smoking cigarettes, alcohol consumption, physical activity, and sleeping, were all found to
have an impact on mental health outcomes. Our study highlights the need for university
administrators and mental health professionals to consider targeted mental health program-
ming for students, particularly for women and those with poor or fair overall perceived
health.

## Introduction

Since the World Health Organization proclaimed a universal pandemic [1], the coronavirus
disease outbreak (COVID-19) has spread globally with a substantial impact on people's lives.
COVID-19 had infected around 480 million individuals as of April 4, 2022, with 6 million
deaths worldwide [2]. With the emergence of the Omicron variant, the number of infections
are increasing exponentially, with around two million people infected per day in the first week
of January 2022 [3]. Since the beginning of the pandemic, many governments have imple-
mented a variety of anti-epidemic measures to contain the spread of the virus, such as restrict-
ing foreign nationals' travel [4, 5], closing public areas, and shutting down entire transit
systems [6]. Preventing the extremely contagious variant from spreading became the world's
top priority [7]. Nonetheless, the prolonged pandemic has raised concerns about the world
population's mental health [8], particularly in relation to the psychological effects of quaran-
tine procedures that disrupted daily routines, such as the suspension of in-person activities,
the adoption of distancing measures, and social isolation. The long duration of associated mea-
sures poses a host of challenges, obstructions, and risks to physical and mental health condi-
tions including depression, anxiety and stress [9, 10]. While the entire population of the world
is impacted, the subgroup of young adults, particularly college students, are thought to be par-
ticularly vulnerable.

Young people's mental health has long been recognized as a global public health concern
[11]. For example, student distress is both an individual and societal challenge, for loss in pro-
ductivity at work and during their study is associated with major economic burdens [12]. Sev-
eral studies have been conducted around the world to investigate the psychological effect of
the COVID-19 pandemic on young people and students' mental health [13–15]. Most studies
discovered increased levels of anxiety, depression, and stress in various countries [16–19].
According to an online cross-sectional multicounty survey of Asian university students (Paki-
stan, China, India, Indonesia, Saudi Arabia, Malaysia and Bangladesh) conducted in 2021,
35.6% expressed mild to severe anxiety [20]. Additionally, Wang et al. 2020 found that 48.14%
of undergraduate and graduate university students in the United States had moderate-to-

severe depression, 38.48% had moderate-to-severe anxiety, and the majority of participants (71.26%) stated their stress levels had increased during the pandemic [21]. Research on university students during the COVID-19 pandemic from Bangladesh, Egypt, Ethiopia, Lebanon, Turkey, and Brazil reported substantial variation in the percentages of depression (21.2% in Ethiopia to 82.4% in Bangladesh), anxiety (27.7% in Ethiopia to 87.7% in Bangladesh), and stress (12.7% in Lebanon to 57.5% in Brazil) symptoms [22–26]. Furthermore, a systematic review and meta-analysis conducted by Wang et al. in 2020 assessing anxiety, depression, and stress prevalence among college students during the COVID-19 pandemic found that the prevalence of anxiety, depression, and stress was 29%, 37%, and 23%, respectively [16].

In Lebanon, the first confirmed COVID-19 case in the country was reported on February 21, 2020. In an attempt to flatten the curve, the government adopted multiple lockdowns between 2020 and 2021, giving authorities the legislative power to implement extraordinary measures against COVID-19, such as border closures (airport, sea, and land) and closures of public and private facilities [27, 28]. The population in Lebanon is around 7 million; since the beginning of the COVID-19 pandemic, the Lebanese Ministry of Public Health (MOPH) has confirmed around one million cases and more than 10 000 death as of April 4, 2022 [29]. To date, 49%, 43% and 24% of the Lebanese population over the age of 12 have received their first, second, and third doses of vaccine, respectively, making the immunization process sluggish [29]. This unstable epidemiological situation, particularly in light of the emergence of a new highly transmissible variant such as Omicron, has given rise to a slew of concerns, including an increase in infection fears and significant lifestyle changes as a result of lockdown measures, all of which have had an impact on the population's psychological well-being and mental health [28]. The negative impact of the COVID-19 pandemic on mental health in Lebanon was demonstrated in tertiary referral hospital population [30], healthcare workers [31, 32], refugees [33], general population [7, 34], and young population (18 to 35 years) [35]. According to a study by Fawaz and Samaha in 2021, 17.9%, 13.8%, and 1.7% of students exhibited mild, moderate, and severe depressive symptoms, respectively; also, mild, moderate, severe, and extremely severe anxiety symptoms were found in 3.3%, 21.9%, 6.3%, and 2.3%, respectively; and 11% of students reported mild stress, while 1.7% reported moderate stress. However, Fawaz and Samaha's study was conducted in April 2020, at the onset of the COVID-19 pandemic, when the impact of the pandemic on mental health was minimal; therefore, preliminary examination of depression, anxiety, and stress suggests that further inquiry on these issues is needed to better document, understand, and plan for appropriate mental health programming for students, especially in light of their increased vulnerability.

This study attempts to help fill the gap in the scarcity of literature on the impact of the pandemic on the mental health of students in Lebanon.

## Materials and methods

### Study design and participants

This cross-sectional survey was conducted in Lebanon using an online survey distributed to undergraduate and graduate university students. A link to the survey, with a study description, was sent to students and faculty via electronic platforms. Data was collected towards the end of the second year of the COVID-19 pandemic, particularly during the Omicron variant emergence (3 November 2021 and 7 February 2022). The sample of this study included 329 students, with the following inclusion criteria: undergraduate and graduate students who were 18 years of age or higher, enrolled at the American University of Beirut (a private university) or the Lebanese University (a public university) between Spring 2020–2021 and Fall 2021–2022.

The questionnaire was distributed in Arabic and English in Lebanon. Prior to filling out the survey, all participants provided written informed consent online. To adapt to the rapidly changing pandemic context and prioritize participant and researcher safety, we employed an online convenience sampling approach. This method was chosen over traditional in-person methods to minimize potential transmission, particularly given the rapid spread associated with the Omicron variant. The decision to use online distribution platforms aligns with previous methodologies adopted in COVID-19-related research [28, 36, 37]. No financial incentive was provided to the participants and anonymity was maintained to ensure the confidentiality and reliability of data. This study was conducted in full compliance with the provisions of the Declaration of Helsinki regarding research on human participants. Ethics approval for the study was obtained from the Institutional Review Board at the American University of Beirut (SBS-2021-0256) and the Research Ethics Board at York University in Canada (Certificate # e2021-327).

## Measures

**Sociodemographic and other factors.** Measured sociodemographic factors were age, gender, income, current program, nationality, relationship status, number of people living in the household. Lifestyle practices included cigarette and shisha smoking, alcohol intake, physical activity, sleeping patterns, internet usage, and overall health. Participants were also asked if they had sought private counseling or therapy from a clinical mental health professional, if they had tried mindfulness meditation, if they had followed COVID-19 preventive measures (wearing masks, handwashing, quarantining, etc.), if they had received a COVID-19 vaccine, and if they had kept up with COVID-19 updates. Finally, participants were asked if they had COVID-19 infection, if they believe that Corona virus and vaccination were the subject of a conspiracy, and if religion is important in their daily lives.

**Mental health outcomes.** *Depression (PHQ-9; Kroenke, 2001).* The Patient Health Questionnaire (PHQ-9) [38] is a brief 9-items, widely used, screening tool that is used to detect depression symptoms in community settings. The Diagnostic and Statistical Manual of Mental Disorders (DSM-IV), 4th Edition, was used to develop the PHQ-9. Prior to administration, each item is assessed for the prior two weeks: 0 = "not at all," 1 = "several days," 2 = "more than half the days," and 3 = "nearly every day," with a total score ranging from 0 to 27, and higher values indicating more severe depression. Minimum depression is indicated by a score of 0–4; mild depression 5–9; moderate depression 10–14; moderately severe depression 15–19; severe depression 20–27 [38]. "Feeling down, depressed, or hopeless," as well as "Poor appetite or overeating," are examples of scale items.

Participants with a score of 10 or above were assigned to the Possible Major Depressive Disorder (MDD) group, while those with a score of 9 or less were assigned to the Non-MDD group [38]. With a sensitivity of 80% and specificity of 92%, a total score of 10 or above indicated the possibility of serious depression [39, 40]. Additionally, PHQ-9 is a self-rating scale with strong reliability and validity for students [41, 42]. The Arabic-translated version of the PHQ-9, which has been validated, demonstrated good reliability with a Cronbach alpha of 0.88 [43]. In our study, the Cronbach's alpha coefficient of the PHQ-9 was 0.901.

*Anxiety (Beck Anxiety Inventory (BAI); Beck et al., 1988).* Anxiety was assessed using the Beck Anxiety Inventory (BAI), a 21-item questionnaire that measures anxiety symptoms [44, 45]. Participants must rate themselves on a 0–3 scale, with zero indicating "Not at all" and three indicating "Severely-It bothered me a lot," with a maximum score of 63 and a minimum score of zero. Minimal anxiety is a score of 0–7, mild anxiety 8–15, moderate anxiety 16–25, and severe anxiety 26–63 [46]. A score of 16 is considered the clinical cut-off for anxiety [47].

The items reflect frequent anxiety symptoms, such as worry of the worse happening, increase in heart rate, fear of losing control, and fear of dying. The BAI demonstrated high internal consistency (Cronbach's alpha = 0.94) and acceptable reliability throughout an average time lapse of 11 days (r = 0.67) in earlier research [48]. The Arabic-translated version of the 21-BAI scale has been validated among university students in Kuwait, with Cronbach's alpha estimated to be between 0.83 and 0.90 [49]. In our study, the Cronbach's alpha coefficient of the BAI scale was 0.944.

*Stress (Perceived Stress Scale (PSS); Cohen, Kamarck & Mermel-stein, 1983).* Generalized stress was measured using the 10-item Perceived Stress Scale (PSS) that measures symptoms of stress [50]. It has negative elements that test lack of control and unpleasant affective reactions, as well as positive elements that examine the ability to cope with current stressors. Item examples include, 'How often have you felt nervous or stressed?' and 'How often have you felt confident about your ability to handle your personal problems?' People rated how often they had experienced these feelings during the past month on a five-point Likert scale from 0 = never to 4 = very often. PSS-10 scores were obtained by reversing the scores on the four positive items; the items were 4, 5, 7 and 8. Total scores vary from 0 to 40, with 0–13 indicating mild stress, 14–26 indicating moderate stress, and 27–40 indicating high stress. In this study, high perceived stress associated with COVID-19 was defined as a score of 27 or above. This cut-off point has been used in a previous study [51].

The PSS is a simple global stress measure that has been proven to be reliable and valid in a variety of settings and languages [52–55]. In particular, the PSS-10 questionnaire was validated to assess stress among university students in a study conducted in China [56]. The Arabic version of the PSS-10 was validated in a study conducted in Lebanon, demonstrating good Cronbach's alpha reliability (0.74) [57]. The Cronbach's alpha coefficient of the PSS-10 scale was 0.846 in this study.

## Data analysis

Descriptive statistics were used to summarize the outcome variables, sociodemographic and other self-reported factors. Continuous variables were summarized as means and standard deviations (SDs), while categorical variables were summarized as frequencies and percentages. The study's dependent variables: depression, anxiety, and stress were dichotomized based on the cut-off points. Three multiple logistic regression models were performed to model the dependent variables using the independent sociodemographic variables and lifestyle factors.

Using simple and multiple logistic regression, the unadjusted and adjusted odds ratios (U-OR; A-OR), as well as the 95% confidence interval (95% CI), were estimated. The Hosmer-Lemeshow test was used to evaluate the logistic models' fit. For the analysis, the R programming language was used (version 4.1.2). The level of statistical significance was set at 0.05.

## Results

### Socio demographic characteristics

The current study included 329 students. Table 1 summarizes the descriptive statistics for the study participants' characteristics. The mean (SD) age of the participants was 24.99 (7.39) years. The majority of participants were females (63.8%). Students were enrolled in a variety of university programs, with undergraduate students accounting for 43% of the sample. More than two-thirds (77.5%) of participants had a household monthly income of 450 dollars or less. Approximately 60% of students considered their overall health to be good, very good, or excellent. Sixty-four percent of the respondents stated that religion is important in their daily lives. Corona virus and vaccination were the subject of a conspiracy, according to 14% of

**Table 1. Socio-demographic characteristics of university students and bivariate relationships (N = 329).**

| | | Depression | | Anxiety | | Stress | |
|---|---|---|---|---|---|---|---|
| **N** | | 329 | | 324 | | 326 | |
| **Mean (SD)** | | 10.18(6.83) | | 18.81(14.42) | | 21.97(7.30) | |
| | **n (%)** | U-OR | **P-value** | U-OR | **P-value** | U-OR | **P-value** |
| **Age (Mean (SD))** | 24.99 (7.39) | 0.94 | **0.001** | 0.97 | **0.04** | 0.97 | 0.139 |
| **Gender** | | | | | | | |
| Men | 77(23.4) | ref | | ref | | ref | |
| women | 210(63.8) | 1.35 | 0.268 | 2.05 | **0.007** | 2.20 | **0.019** |
| *Missing* | 42(12.8) | | | | | | |
| **Relationship status** | | | | | | | |
| Not in a relationship | 172(52.3) | 1.51 | 0.088 | 1.15 | 0.553 | 1.35 | 0.267 |
| In a relationship | 118(35.9) | ref | | ref | | ref | |
| *Missing* | 39(11.9) | | | | | | |
| **University program** | | | | | | | |
| Undergraduate degree | 143(43.5) | 2.12 | **0.001** | 1.62 | **0.044** | 1.77 | **0.031** |
| Certificate program | 14(4.3) | 4.55 | **0.01** | 3.93 | **0.04** | 2.55 | 0.103 |
| Graduate program (MA or MSc) | 141(42.9) | ref | | ref | | ref | |
| PhD Program | 12(3.6) | 0.91 | 0.88 | 0.21 | 0.05 | 2.17 | 0.982 |
| MD program | 19(5.8) | 1.32 | 0.57 | 0.78 | 0.616 | 1.572 | 0.395 |
| *Missing* | | | | | | | |
| **GPA status during the pandemic** | | | | | | | |
| No change | 103(31.3) | 0.78 | 0.373 | 0.78 | 0.375 | 0.71 | 0.281 |
| Decreased | 103(31.3) | 1.36 | 0.248 | 1.07 | 0.782 | 0.81 | 0.471 |
| Increased | 123(37.4) | ref | | ref | | ref | |
| **Income (in USD)** | | | | | | | |
| ≤450 | 255(77.5) | 1.28 | 0.352 | 2.00 | **0.010** | 1.03 | 0.899 |
| >450 | 74(22.5) | ref | | ref | | ref | |
| **Overall rated health** | | | | | | | |
| Poor | 29(8.8) | 74.99 | **<0.001** | 335.90 | **<0.001** | 2.97 | 0.979 |
| Fair | 100(30.4) | 23.29 | **0.003** | 30.85 | **0.001** | 1.04 | 0.981 |
| Good | 121(36.8) | 6.37 | **0.07** | 7.89 | 0.050 | 4.07 | 0.982 |
| Very good | 66(20.1) | 3.52 | 0.24 | 7.80 | 0.055 | 2.47 | 0.982 |
| Excellent | 13(4.0) | ref | | ref | | ref | |
| **Importance of religion in daily decisions** | | | | | | | |
| Not important | 70(21.3) | ref | | ref | | | |
| Important | 213(64.7) | 0.67 | 0.160 | 1.10 | 0.709 | 0.61 | 0.095 |
| *Missing* | 46(14.0) | | | | | | |
| **Conspiracy behind COVID virus/vaccine** | | | | | | | |
| Disapprove | 117(35.6) | 0.56 | 0.098 | 0.88 | 0.727 | 0.71 | 0.358 |
| Neither approve nor disapprove | 117(35.6) | 0.69 | 0.293 | 0.95 | 0.882 | 0.81 | 0.566 |
| Approve | 49(14.9) | ref | | ref | | | |
| *Missing* | 46(14.0) | | | | | | |
| **Adherence to COVID-19 preventive measures** | | | | | | | |
| No | 41(12.5) | 2.07 | **0.0348** | 1.62 | 0.165 | 0.80 | 0.587 |
| Yes | 242(73.6) | ref | | ref | | | |
| *Missing* | 46(14.0) | | | | | | |
| **Infected with COVID-19** | | | | | | | |
| No | 197(59.9) | 1.01 | 0.958 | 1.29 | 0.314 | 0.92 | 0.774 |

*(Continued)*

**Table 1.** (Continued)

|  |  | Depression |  | Anxiety |  | Stress |  |
|---|---|---|---|---|---|---|---|
| Yes | 86(26.1) | ref |  | ref |  | ref |  |
| *Missing* | 46(14.0) |  |  |  |  |  |  |
| **Private Counseling** |  |  |  |  |  |  |  |
| No | 140 (42.6) | ref |  | ref |  | ref |  |
| Yes | 189 (57.4) | 1.24 | 0.06 | 1.77 | **0.010** | 2.09 | **0.004** |

U-OR depicts the unadjusted bivariate odds ratio.

participants. Furthermore, the majority of students (73.6%) followed COVID-19 prevention guidelines, and about a quarter of them were infected with COVID-19. Private counseling was received by more than half of the students (57.4%).

## Lifestyle practices

Regarding lifestyle practices during the pandemic (Table 2), around two-thirds (63.5%) of the participants followed a healthy diet. An increase in cigarette and shisha smoking, as well as alcohol usage, was self-reported by around 12% of the respondents. Physical activity decreased for nearly half of the students, while it increased for 31%. A third of the participants (32.2%) slept for fewer than seven hours, while 17.3% slept for more than nine hours. The majority of participants (70%) used the internet for at least 3 hours daily.

## Mental health outcomes

The mental health outcome scales were shown to have strong internal consistency (0.901 for depression, 0.944 for anxiety, and 0.846 for stress) in the study sample, as determined by Cronbach alpha. The mean (SD) score for depression was 10.18 (6.83), anxiety was 18.81 (14.42), and stress was 21.97 (7.30). Fig 1 depicts the study participants' rates of depression, anxiety, and stress. Mild to moderate depression, anxiety, and stress were reported by the majority of participants (52.3%, 42.9%, and 61.7%, respectively), while severe depression, severe anxiety, and high stress were reported by 24.6%, 29.3%, and 27.6%, respectively. In total, students reported moderate to severe rates of depression, anxiety, and stress at a rate of 75.9%, 72.2%, and 89.3% respectively.

Fig 2 highlights students' lifestyle practices during the pandemic, whereas Fig 3 shows changes in lifestyle practices during the pandemic in comparison to before the pandemic. A significant difference in internet use, sleeping hours, and following a healthy diet was noted among university students during the pandemic (p-value = 0.001) as compared to before the pandemic. Tables 1 and 2 demonstrate the findings of the simple logistic regression analysis.

According to the results of multiple logistic regression analysis (Table 3) and after adjusting for sociodemographic and lifestyle factors, women have nearly three times the odds of anxiety and stress as men (AOR anxiety = 2.69, 95% CI: 1.20–5.75; p-value = 0.011; AOR stress = 2.93, 95% CI: 1.12–7.65; p-value = 0.028). University programs were linked to anxiety and stress. PhD students reported lower levels of anxiety than undergraduates (AOR anxiety/PhD program = 0.06, 95% CI: 0.01–0.71, p-value = 0.033).

Furthermore, depression and anxiety were linked to overall health rating, where the odds of depression are 30 and 418 times higher among those who rate their health as poor and fair, respectively, than among those who rate their health as excellent (AOR Depression/poor health = 30.59, 95% CI:2.44–384.00, p-value = 0.008; AOR Depression/Fair health = 14.12,

**Table 2. Association between university students' mental health outcomes and lifestyle practices during the pandemic and bivariate relationships.**

| | n (%) | Depression | | Anxiety | | Stress | |
|---|---|---|---|---|---|---|---|
| | | U-OR | P-value | U-OR | P-value | U-OR | P-value |
| **Follow healthy diet** | | | | | | | |
| No | 120(36.5) | 2.18 | **<0.001** | 1.46 | 0.100 | 2.75 | **<0.001** |
| Yes | 209(63.5) | ref | | ref | | ref | |
| **Cigarette smoking** | | | | | | | |
| No practice | 279(84.8) | ref | | ref | | ref | |
| Reduced | 8(2.4) | 0.81 | 0.786 | 3.02 | 0.180 | 1.69 | 0.47 |
| Increased | 42(12.8) | 2.72 | **0.004** | 2.51 | **0.010** | 1.91 | **0.05** |
| **Shisha smoking** | | | | | | | |
| No practice | 262(79.6) | ref | | ref | | ref | |
| Reduced | 27(8.2) | 0.62 | 0.277 | 0.96 | 0.935 | 0.56 | 0.269 |
| Increased | 40(12.2) | 1.88 | 0.066 | 1.21 | 0.569 | 1.20 | 0.616 |
| **Alcohol consumption** | | | | | | | |
| No practice | 252(76.6) | ref | | ref | | ref | |
| Reduced | 38(11.6) | 0.86 | 0.684 | 0.55 | 0.090 | 1.03 | 0.921 |
| Increased | 39(11.9) | 1.02 | 0.952 | 1.36 | 0.370 | 1.00 | 0.997 |
| **Physical activity** | | | | | | | |
| No practice | 63(19.1) | 2.50 | **0.005** | 1.38 | 0.311 | 2.68 | **0.008** |
| Reduced | 164(49.8) | 1.90 | **0.013** | 1.23 | 0.405 | 2.45 | **0.004** |
| Increased | 102(31.0) | ref | | ref | | ref | |
| **Sleeping hours** | | | | | | | |
| <7 | 106(32.2) | 2.46 | **<0.001** | 1.77 | **0.023** | 1.32 | 0.315 |
| 7–9 | 166(50.5) | ref | | ref | 0.726 | ref | |
| >9 | 57(17.3) | 2.90 | **<0.001** | 1.11 | | 2.63 | **0.002** |
| **Internet use (in hours)** | | | | | | | |
| <1 | 12(3.6) | ref | | ref | | ref | |
| [1–2[ | 31(9.4) | 1.26 | 0.744 | 1.58 | 0.502 | 0.69 | 0.623 |
| [2–3[ | 59(17.9) | 0.95 | 0.939 | 0.68 | 0.553 | 0.360 | 0.151 |
| [3–4[ | 57(17.3) | 1.35 | 0.652 | 1.28 | 0.698 | 0.47 | 0.291 |
| ≥4 | 170(51.7) | 2.30 | 0.186 | 1.23 | 0.722 | 1.11 | 0.859 |

U-OR depicts the unadjusted bivariate odds ratio

95% CI: 1.49–133.69, p-value = 0.021). When compared to those who rated their health as excellent, the odds of anxiety are 418 and 21 times higher for those who rated their health as poor and fair, respectively (AOR Anxiety/poor health = 418.87, 95% CI: 18.97–9251.40, p-value = <0.001; AOR Anxiety/Fair health = 21.85, 95% CI: 2.26–211.04, p-value = 0.008). Students who have said religion is important in their daily decisions also reported less stress than those who said religion is not important in their daily decisions (AOR Stress/Important = 0.38, 95%CI: 0.18–0.97, p-value = 0.043).

Additionally, stress was associated with private counseling, with those who sought private counseling having four-fold higher odds of stress than those who did not (AOR Stress = 4.37, 95% CI:1.96–9.76, p-value = <0.001). Cigarette smoking was associated with all mental health outcomes where individuals who increased their smoking intake during the pandemic, had nearly three time the odds of depression and four times the odds of anxiety and stress compared to those who did not smoke (AOR Depression = 2.91, 95%CI: 1.10–7.73, p-value = 0.032; AOR Anxiety = 4.26, 95% CI:1.57–11.53, p-value = 0.004; AOR Stress = 4.66,

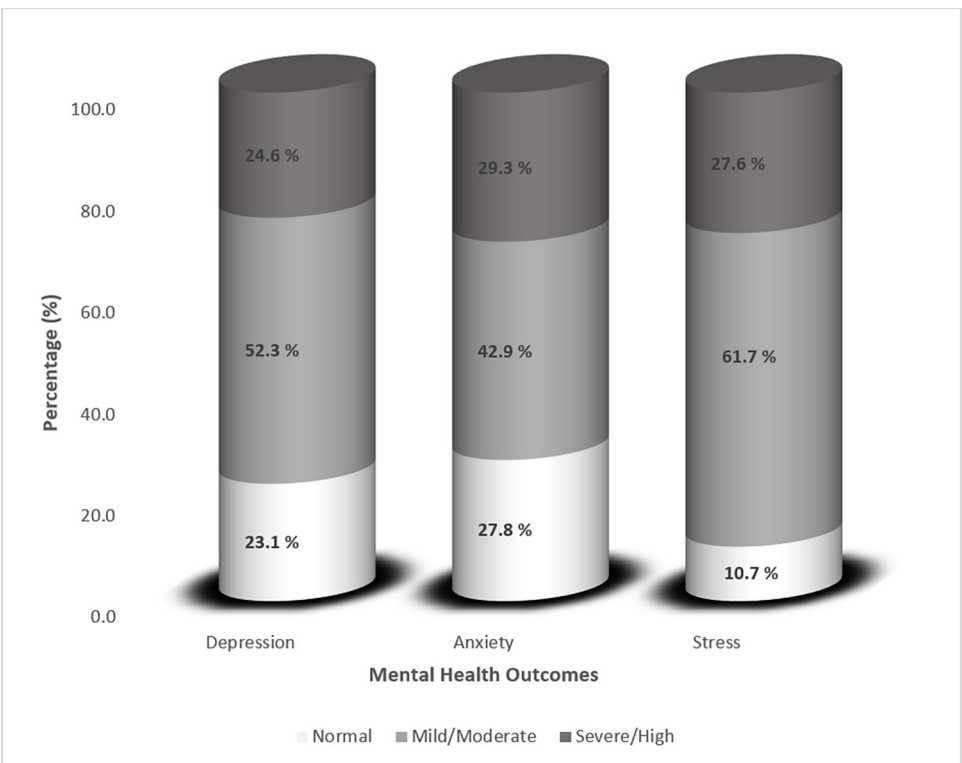

**Fig 1. Depression, anxiety and stress levels among university students in Lebanon.**

95% CI: 1.60–13.56, p-value = 0.005). Students who reduced their alcohol consumption have lower odds of anxiety compared to those who do not consume alcohol (AOR anxiety = 0.22, 95%CI:0.06–0.85, p-value = 0.028). The odds of stress are two times higher among those who reduced their physical activity than in students who increased it (AOR Stress = 2.42, 95% CI: 1.05–5.59, p-value = 0.038). Students who slept fewer than seven hours per day had nearly three times the risk of depression than those who sleep seven to nine hours (AOR depression = 3.19, 95% CI: 1.54–6.61, p-value = 0.002). Finally, the Hosmer-Lemeshow p-values for the final models of depression, anxiety, and stress were 0.517, 0.893, 0.496, respectively, all of which are greater than 0.05, indicating adequate model fit.

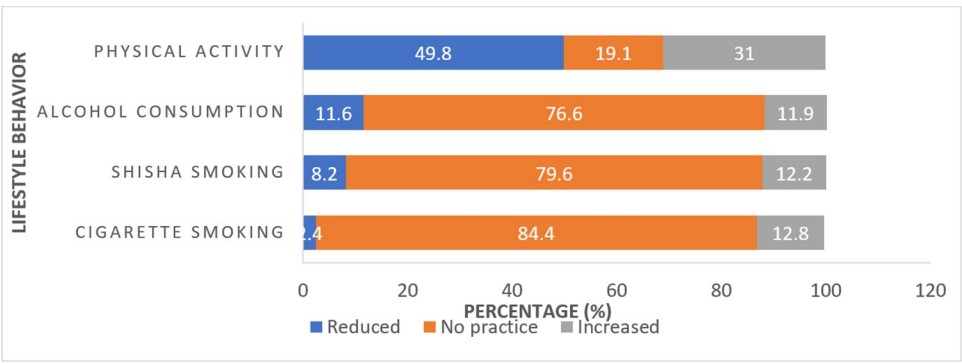

**Fig 2. Lifestyle practices during the pandemic.**

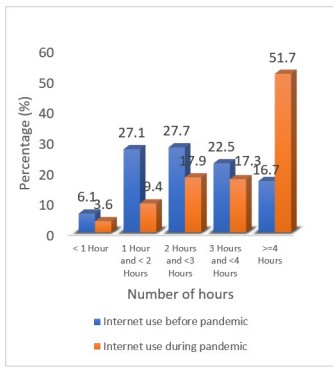
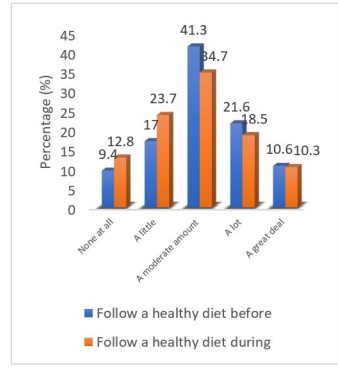
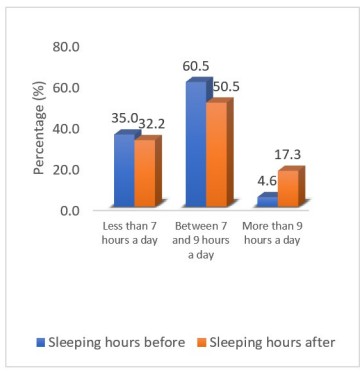

a. Internet use before and during the pandemic
(McNemar-Bowker Test=155.10, df=9, p <0.001)

b. Following a healthy diet before and during the pande
(McNemar-Bowker Test=17.10, df=9, p=0.047)

c. Sleeping hours before and during the pandemic
(McNemar-Bowker Test=27.65, df=3, p-value<0.001)

**Fig 3. Changes in lifestyle practices.**

All mental health outcomes had no significant association with the importance of religion in everyday decisions, adherence to COVID-19 measures, being previously infected with COVID-19, or internet use.

## Discussion

There is currently a scarcity of studies assessing the mental health of university students in Lebanon. This study aimed at understanding university students' mental health, specifically depression, anxiety, and stress, during Lebanon's extended COVID-19 pandemic, as well as the sociodemographic factors and lifestyle practices associated with it. University students are frequently regarded as the backbone of society, and their mental health during a pandemic may significantly impact their performance and life trajectories. Our results indicate that gender, university program, overall rated health, importance of religion in daily decisions, private counseling, smoking cigarettes, alcohol consumption, physical activity, and sleeping, are factors that influence these mental health outcomes.

In our study, moderate to severe rates of depression, anxiety, and stress were higher 2.2 times for depression, 2.3 times for anxiety, and 7 times for stress than previously reported rates among university students in Lebanon in April 2020 by Fawaz and Samaha in 2021. This might be explained by the increased economic downturn of the country where the devaluation of the money has almost doubled and it is known from pre-pandemic studies that depression, anxiety and stress have been associated with financial crisis [58, 59]. Another contributing factor might be the explosion of the Beirut port that destroyed a large area of the capital and impacted directly thousands of families with documented devastating mental health impact [60].

### Gender

Our findings point to a gender difference in symptoms of anxiety and stress, such as the odds of anxiety and stress were 2.7 and nearly 3 times higher among women compared to men. This is congruent with findings in other studies where being a female was found to be a risk factor for poor mental health among students [61–65].

### Self-rated health

Importantly, overall rated health stood out as the strongest predictor of depression and anxiety. Depression and anxiety were 30 times and 418 times, respectively, as high in the "poor"

**Table 3. Adjusted multiple logistic regression analyses for the depression, anxiety and stress outcomes.**

| | Depression | | | Anxiety | | | Stress | | |
|---|---|---|---|---|---|---|---|---|---|
| | A-OR | 95% CI | P-value | A-OR | 95% CI | P-value | A-OR | 95% CI | P-value |
| **Age** | 0.94 | 0.89–1.00 | 0.063 | 1.00 | 0.95–1.07 | 0.875 | 1.01 | 0.94–1.08 | 0.779 |
| **Gender** | | | | | | | | | |
| Men | ref | | | ref | | | ref | | |
| women | 1.61 | 0.75–3.45 | 0.222 | 2.69 | 1.20–5.75 | **0.011** | 2.93 | 1.12–7.65 | **0.028** |
| **Relationship status** | | | | | | | | | |
| Not in a relationship | 1.08 | 0.85–1.37 | 0.528 | 1.12 | 0.88–1.43 | 0.339 | 1.17 | 0.90–1.53 | 0.249 |
| In a relationship | ref | | | ref | | | ref | | |
| **University program** | | | | | | | | | |
| Undergraduate degree | 1.54 | 0.73–3.24 | 0.259 | 1.94 | 0.93–4.08 | 0.079 | 2.28 | 0.95–5.49 | 0.067 |
| Certificate program | 3.12 | 0.56–17.44 | 0.195 | 3.35 | 0.52–21.66 | 0.204 | 2.87 | 0.53–15.65 | 0.224 |
| Graduate program (MA or MSc) | ref | | | ref | | | ref | | |
| PhD Program | 1.40 | 0.26–7.50 | 0.692 | 0.06 | 0.01–0.71 | **0.026** | 0.00 | 0.00-Inf | 0.992 |
| MD program | 1.10 | 0.20–6.11 | 0.912 | 0.86 | 0.16–4.66 | 0.861 | 2.68 | 0.42–17.03 | 0.297 |
| **GPA status during the pandemic** | | | | | | | | | |
| No change | 1.14 | 0.54–2.43 | 0.729 | 1.48 | 0.68–3.22 | 0.322 | 0.88 | 0.36–2.11 | 0.768 |
| Decreased | 1.40 | 0.69–2.87 | 0.352 | 1.34 | 0.64–2.81 | 0.436 | 0.92 | 0.41–2.03 | 0.830 |
| Increased | ref | | | ref | | | ref | | |
| **Income (in USD)** | | | | | | | | | |
| ≤450 | 0.85 | 0.40–1.82 | 0.675 | 2.00 | 0.93–4.30 | 0.078 | 0.94 | 0.38–2.32 | 0.885 |
| >450 | ref | | | ref | | | ref | | |
| **Overall rated health** | | | | | | | | | |
| Poor | 30.59 | 2.44–384.00 | **0.008** | 418.87 | 18.97–9251.40 | **<0.001** | 147067582.78 | 0.00-Inf | 0.991 |
| Fair | 14.12 | 1.49–133.69 | **0.021** | 21.85 | 2.26–211.04 | **0.008** | 44692700.22 | 0.00-Inf | 0.992 |
| Good | 4.42 | 0.49–40.27 | 0.187 | 4.47 | 0.48–41.43 | 0.187 | 16710160.79 | 0.00-Inf | 0.992 |
| Very good | 3.17 | 0.33–30.14 | 0.315 | 7.67 | 0.81–72.92 | 0.076 | 16853535.43 | 0.00-Inf | 0.992 |
| Excellent | ref | | | ref | | | ref | | |
| **Importance of religion in daily decisions** | | | | | | | | | |
| Not important | ref | | | ref | | | ref | | |
| Important | 0.46 | 0.20–1.10 | 0.080 | 1.06 | 0.44–2.52 | 0.904 | 0.38 | 0.15–0.97 | **0.043** |
| **Conspiracy behind COVID virus/vaccine** | | | | | | | | | |
| Disapprove | 0.65 | 0.26–1.60 | 0.346 | 1.42 | 0.56–3.63 | 0.461 | 0.53 | 0.18–1.52 | 0.235 |
| Neither approve nor disapprove | 1.00 | 0.41–2.44 | 0.993 | 1.68 | 0.66–4.28 | 0.275 | 0.83 | 0.31–2.23 | 0.706 |
| Approve | ref | | | ref | | | ref | | |
| **Adherence to COVID-19 preventive measures** | | | | | | | | | |
| No | 1.22 | 0.52–2.89 | 0.643 | 1.34 | 0.55–3.24 | 0.516 | 0.35 | 0.12–1.01 | 0.051 |
| Yes | ref | | | ref | | | ref | | |
| **Infected with COVID-19** | | | | | | | | | |
| No | 0.80 | 0.40–1.58 | 0.515 | 1.25 | 0.64–2.46 | 0.512 | 0.84 | 0.40–1.76 | 0.640 |
| Yes | ref | | | ref | | | ref | | |
| **Private Counseling** | | | | | | | | | |
| No | ref | | | ref | | | ref | | |
| Yes | 1.29 | 0.68–2.46 | 0.433 | 1.60 | 0.83–3.08 | 0.160 | 4.37 | 1.96–9.76 | **<0.001** |
| **Follow healthy diet** | | | | | | | | | |
| No | 1.27 | 0.65–2.49 | 0.492 | 0.91 | 0.45–1.82 | 0.784 | 1.82 | 0.86–3.85 | 0.117 |
| Yes | ref | | | ref | | | ref | | |
| **Cigarette smoking** | | | | | | | | | |

*(Continued)*

**Table 3.** (Continued)

| | Depression | | | Anxiety | | | Stress | | |
|---|---|---|---|---|---|---|---|---|---|
| | A-OR | 95% CI | P-value | A-OR | 95% CI | P-value | A-OR | 95% CI | P-value |
| No practice | ref | | | ref | | | ref | | |
| Reduced | 0.73 | 0.10–5.43 | 0.759 | 6.73 | 0.67–67.71 | 0.105 | 1.77 | 0.24–13.15 | 0.578 |
| Increased | 2.91 | 1.10–7.73 | **0.032** | 4.26 | 1.57–11.53 | **0.004** | 4.66 | 1.60–13.56 | **0.005** |
| **Shisha smoking** | | | | | | | | | |
| No practice | ref | | | ref | | | ref | | |
| Reduced | 0.75 | 0.25–2.30 | 0.618 | 0.83 | 0.28–2.41 | 0.728 | 0.97 | 0.24–3.88 | 0.964 |
| Increased | 1.00 | 0.40–2.53 | 0.994 | 0.39 | 0.14–1.10 | 0.075 | 0.48 | 0.17–1.37 | 0.169 |
| **Alcohol consumption** | | | | | | | | | |
| No practice | ref | | | ref | | | ref | | |
| Reduced | 0.52 | 0.16–1.70 | 0.278 | 0.22 | 0.06–0.85 | **0.028** | 0.24 | 0.06–1.02 | 0.054 |
| Increased | 0.43 | 0.14–1.27 | 0.127 | 0.96 | 0.33–2.82 | 0.943 | 0.29 | 0.08–1.05 | 0.060 |
| **Physical activity** | | | | | | | | | |
| No practice | 1.95 | 0.75–5.04 | 0.170 | 0.94 | 0.36–2.41 | 0.890 | 1.70 | 0.61–4.72 | 0.310 |
| Reduced | 1.85 | 0.90–3.80 | 0.095 | 1.29 | 0.62–2.65 | 0.496 | 2.42 | 1.05–5.59 | **0.038** |
| Increased | ref | | | ref | | | ref | | |
| **Sleeping hours** | | | | | | | | | |
| <7 | 3.19 | 1.54–6.61 | **0.002** | 1.65 | 0.80–3.40 | 0.174 | 0.87 | 0.38–1.99 | 0.744 |
| 7 to 9 | ref | | | ref | | | ref | | |
| >9 | 1.65 | 0.70–3.89 | 0.254 | 0.53 | 0.22–1.28 | 0.158 | 0.69 | 0.26–1.88 | 0.473 |
| **Internet use (in hours)** | | | | | | | | | |
| <1 | ref | | | ref | | | ref | | |
| [1–2[ | 1.17 | 0.17–8.13 | 0.875 | 1.53 | 0.21–10.93 | 0.670 | 0.61 | 0.08–4.84 | 0.643 |
| [2–3[ | 1.08 | 0.18–6.55 | 0.930 | 0.76 | 0.13–4.54 | 0.759 | 0.25 | 0.04–1.74 | 0.163 |
| [3–4[ | 2.27 | 0.36–14.29 | 0.382 | 1.55 | 0.24–9.79 | 0.644 | 0.55 | 0.08–3.87 | 0.546 |
| ≥4 | 2.99 | 0.53–16.91 | 0.216 | 1.27 | 0.23–7.13 | 0.787 | 1.24 | 0.21–7.38 | 0.812 |

A-OR depicts the adjusted odds ratio of a model including each outcome with all of the sociodemographic factors and lifestyle practices.

overall health group as those in the "excellent" group. While depression and anxiety were 14 times and 21 times as high in the "fair" overall health group as those in the "excellent" group. This is in line with other studies that uncovered that poor overall health was among the strongest predictors of these outcomes of depression and anxiety [66].

## Lifestyle practices

Nearly half of the students reported a decrease in physical activity, and this translated into a nearly 2.5 times increase in experienced stress among that group. Those results are consistent with other findings documenting a global shift in certain lifestyle practices. For example, the total physical activity in Italy fell dramatically during the first COVID-19 wave compared to before across all age categories, particularly in men [67]; however, changes in physical activities vary among countries. Studies from Belgium, France, and Switzerland, for instance, have found an overall rise in both physical activity [68, 69] and sedentary behavior [68]. This may be related to differences in cultural norms and/or governmental policies and investments that may support or hinder opportunities for physical activities during the pandemic; the European union has been known to take special care to implement policies promoting physical activity during the COVID-19 pandemic [70].

In terms of alcohol and smoking habits, this study shows an overall increase in cigarette and shisha smoking, as well as alcohol consumption during the COVID-19 pandemic. While shisha smoking had no effect on mental health outcomes, the increase in cigarette smoking worsened mental health outcomes for those who smoke by increasing the risk for depression, anxiety, and stress by nearly 3 times, 4 times, and 4.5 times, respectively, compared to non-smokers; this is in line with studies that found a positive association between smoking exposure and depression or anxiety [71]. Reduced alcohol consumption, on the other hand, was associated with significant reductions in anxiety. Existing research on alcohol consumption and smoking is mixed. Some studies have found no difference in alcohol intake during home confinement [72, 73] and a decrease in smoking [73, 74], while others found an increase in both alcohol consumption [75, 76] and smoking [75, 77]. However, it is worth noting that these studies reflect the situation at the early stages of the COVID-19 pandemic while our study was conducted two years afterwards.

Also, the odds of depression among students who slept less than 7 hours a day, were 3 times higher than in those who slept 7 to 9 hours a day; this is consistent with meta analysis studies that showed that sleep difficulty was significantly associated with depression [78, 79].

It is worthy to note that our study shows a significant decrease in following a healthy diet among university students during the pandemic as compared to before the pandemic; however, that was not translated with an impact on mental health. The literature is mixed as some studies have found minor changes in dietary behaviors [72, 80], while others depict an increase in unhealthy food consumption, overeating, and snacking in between meals [72, 73, 81].

## Religion and private counseling

Several studies found an association between the practice of religion and lower stress levels [82, 83], including during the COVID-19 pandemic [84]. In our study, the importance of religion in daily decisions was found to be significantly associated with a significant decrease in stress among study participants. Given the important role religion appears to play in Lebanese society [85], universities should take this into account while developing mental health programs. Furthermore, our findings imply that students who sought private counseling services were twice as likely as those who did not use private counseling services to experience worry and stress.

## Income

Finally, several studies have indicated that lower income contributes to poor mental health [64, 65] and that lower socioeconomic status has been linked to worsening in mental health [86, 87], including higher mortality and suicide rates, which are associated with economic downturns [88]. Our findings revealed no significant association between income and mental health outcomes, which is surprising given the well-established literature related to the social determinants of health. This could be due to the lack of variation in income among participants, as all local students were affected by the 90% local currency devaluation [89] and the fact that 85% of the respondents had an income of less than USD 1,000.

## Limitations of the study

This study has several limitations. First, given the cross-sectional nature of the study design, the results are subject to confounding biases such as the participants' mental health status prior to the COVID-19 pandemic and other life stressors (e.g., experiences of violence). Second, there is the possibility of selection bias as participation was voluntary. Third, the study relied on a convenience sample limited to students from two universities. While this sampling

technique does not necessarily assure that results are generalizable, it can be a useful tool for determining the likelihood of a potential relationship between the variables [90, 91]. Lastly, like any research conducted in an unstable environment with insecurity and instability, as well as constantly changing circumstances, predicting, and isolating the impact of these life factors is nearly impossible.

## Conclusion

We conducted a cross-sectional study among students from two universities in Lebanon. Our data reveal a national student mental health crisis, with exceptionally high rates of moderate to severe depression, anxiety, and stress. Our research also highlights the need of university administrators and mental health specialists paying close attention to the unique needs of female students, as well as those who have a poor or fair self-perceived health, as they are disproportionately affected by mental health issues. Specific programs addressing these categories are essential for their mental health, academic development, and economic contribution. Policies promoting physical activity may be crucial to develop when addressing mental health programming, and the role of religion in a student's life may be a component to consider.

## Acknowledgments

We would like to thank the Canadian Lebanese Academic Forum for facilitating the team building effort.

## Author Contributions

**Conceptualization:** Reem Hoteit, Imad Bou-Hamad, Sahar Hijazi, Dinah Ayna, Maya Romani, Christo El Morr.

**Data curation:** Reem Hoteit, Christo El Morr.

**Formal analysis:** Reem Hoteit.

**Investigation:** Reem Hoteit.

**Methodology:** Reem Hoteit, Imad Bou-Hamad, Sahar Hijazi, Dinah Ayna, Maya Romani, Christo El Morr.

**Project administration:** Reem Hoteit, Christo El Morr.

**Supervision:** Christo El Morr.

**Validation:** Christo El Morr.

**Writing – original draft:** Reem Hoteit, Christo El Morr.

**Writing – review & editing:** Reem Hoteit, Imad Bou-Hamad, Sahar Hijazi, Dinah Ayna, Maya Romani, Christo El Morr.

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
