## [Decision Letter · Decision Letter 0]

26 Feb 2024

PONE-D-23-37343A Cross-sectional Study of University Students' Mental Health and Lifestyle practices Amidst the COVID-19 PandemicPLOS ONE

Dear Dr. Hoteit,

Thank you for submitting your manuscript to PLOS ONE. After careful consideration, we feel that it has merit but does not fully meet PLOS ONE’s publication criteria as it currently stands. Therefore, we invite you to submit a revised version of the manuscript that addresses the points raised during the review process.

We look forward to receiving your revised manuscript.

Kind regards,

Amin Nakhostin-Ansari

Academic Editor

PLOS ONE

2. We noted in your submission details that a portion of your manuscript may have been presented or published elsewhere. [Data used in this manuscript have been deposited in a data bank maintained by The American University of Beirut’s libraries.

We would like to clarify that the data was used in another manuscript published in Plos One (https://doi.org/10.1371/journal.pone.0288358). There could be an overlap in the materials and methods section, however the overall paper tackles a different research question] Please clarify whether this publication was peer-reviewed and formally published. If this work was previously peer-reviewed and published, in the cover letter please provide the reason that this work does not constitute dual publication and should be included in the current manuscript.

Reviewers' comments:

Reviewer's Responses to Questions

**Comments to the Author**

1. Is the manuscript technically sound, and do the data support the conclusions?

Reviewer #1: Yes

Reviewer #2: Yes

2. Has the statistical analysis been performed appropriately and rigorously? 

Reviewer #1: Yes

Reviewer #2: Yes

3. Have the authors made all data underlying the findings in their manuscript fully available?

Reviewer #1: Yes

Reviewer #2: Yes

4. Is the manuscript presented in an intelligible fashion and written in standard English?

Reviewer #1: Yes

Reviewer #2: Yes

5. Review Comments to the Author

Reviewer #1: The manuscript addresses a topic that is currently widely discussed, which is the impacts of the Covid-19 pandemic on the mental health of young adults, particularly university students.

The authors carried out a cross-sectional study and used the strategy of recruiting participants through social networks, asking them to voluntarily respond to a Forms questionnaire, in their own homes.

Evaluating the body of the manuscript, it is observed that the Abstract does not adequately inform, even succinctly, the Objective, Methods and Conclusion.

Objective: the paragraph is a justification. To be Objective, it must contain a verb that expresses a verifiable action; Method: it must minimally contain information on which instruments were used; Conclusion: it must directly answer the Objective.

In the body of the manuscript, in Method, it is necessary for the authors to explain how, once data collection was carried out through a Forms questionnaire, the option for a larger sample would pose a risk of spreading the Covid-19 virus.

Still in Method, it is necessary for authors to justify the choice of the PHQ-9, BAI and PSS instruments and not others, with the same purpose. It is also necessary to provide information on whether the instruments used were validated in the language, in addition to English, which was used by the authors.

After these adjustments, the manuscript must be reevaluated.

Reviewer #2: The submitted paper is a cross-sectional study of depression, anxiety, stress, and life factors in students during the COVID-19 pandemic, and the paper contains many important findings that will be useful to readers.

I have no opinion on major revisions to the content of the text, but the inclusion of references in the text needs to be corrected. Specifically, the year of publication needs to be added to the references in the text, as sometimes only the author's name is given and not the year of publication. For example, the article by Maalouf et al. and the article by Rogowska et al.

Overall, the resolution of the figures is poor and I do not think they are of publishable quality. The resolution of the figures needs to be improved.

6. PLOS authors have the option to publish the peer review history of their article (what does this mean?). If published, this will include your full peer review and any attached files.

Reviewer #1: **Yes: **Rosana Cipolotti

Reviewer #2: No

---

## [Author Response · Author response to Decision Letter 0]

11 Mar 2024

10-March-2024

PLOS ONE

Re Manuscript ID: PONE-D-23-37343 entitled “A Cross-sectional Study of University Students' Mental Health and Lifestyle practices Amidst the COVID-19 Pandemic.”

We would like to thank the editor and reviewers for the opportunity to review our manuscript entitled: “A Cross-sectional Study of University Students' Mental Health and Lifestyle practices Amidst the COVID-19 Pandemic.”

We deeply appreciate all the helpful feedback and suggestions that helped us enhance the article's content. We responded to all of the journal’s and reviewers’ comments and made significant improvements.

Our responses (in bold black and blue, with blue indicating newly included or modified statements in the paper) are described below in a point-to-point manner.

Authors’ response: 

Please note that that the manuscript format has been updated to match the requirements of PLOS ONE.

2. We noted in your submission details that a portion of your manuscript may have been presented or published elsewhere. [Data used in this manuscript have been deposited in a data bank maintained by The American University of Beirut’s libraries. We would like to clarify that the data was used in another manuscript published in Plos One (https://doi.org/10.1371/journal.pone.0288358). There could be an overlap in the materials and methods section, however the overall paper tackles a different research question] .

Please clarify whether this publication was peer-reviewed and formally published. If this work was previously peer-reviewed and published, in the cover letter please provide the reason that this work does not constitute dual publication and should be included in the current manuscript.

Authors’ response: 

Regarding the previous publication entitled "Coping with the COVID-19 pandemic: A cross-sectional study to investigate how mental health, lifestyle, and socio-demographic factors shape students’ quality of life", we confirm that this work underwent peer review and was formally published in PLOS ONE https://doi.org/10.1371/journal.pone.0288358

We assure you that while there may be some overlap in the materials and methods section, the overall focus and research questions addressed in the current study differ significantly from those in the previous publication. The previous study aimed to investigate the social, lifestyle, and mental health aspects associated with quality of life among university students in Lebanon. Whereas, our current study focuses on understanding university students’ mental health, specifically depression, anxiety, and stress, during Lebanon's extended COVID-19 pandemic, as well as the sociodemographic factors and lifestyle practices associated with it. We believe that these distinct research questions warrant separate publications and contribute unique insights to the existing literature. We appreciate your consideration and assure you of our commitment to academic integrity.

3. Please include captions for your Supporting Information files at the end of your manuscript, and update any in-text citations to match accordingly. Please see our Supporting Information guidelines for more information: http://journals.plos.org/plosone/s/supporting-information . 

Authors’ response: 

Thank you. We would like to clarify that our manuscript does not contain any supporting information. The manuscript includes three tables and three figures; with figure captions located at the end of the manuscript. These figures were already submitted along with the manuscript through the submission portal. We appreciate your understanding.

Reviewer #1: 

The manuscript addresses a topic that is currently widely discussed, which is the impacts of the Covid-19 pandemic on the mental health of young adults, particularly university students.

The authors carried out a cross-sectional study and used the strategy of recruiting participants through social networks, asking them to voluntarily respond to a Forms questionnaire, in their own homes.

1. Evaluating the body of the manuscript, it is observed that the Abstract does not adequately inform, even succinctly, the Objective, Methods and Conclusion.

Objective: the paragraph is a justification. To be Objective, it must contain a verb that expresses a verifiable action; Method: it must minimally contain information on which instruments were used; Conclusion: it must directly answer the Objective.

Authors’ response: 

Thank you for your feedback. We have made revisions to the abstract to better align with your suggestions. Specifically, we have ensured that the Objective now includes a clear action verb to express the research aim. In the Methods section, we have included details about the instruments used in the study. Furthermore, the Conclusion has been refined to directly address the research objective. We appreciate your detailed review and hope that these changes improve the clarity and coherence of the abstract.

Objectives

University students are regarded as the backbone of society, and their mental health during a pandemic may have a substantial impact on their performance and life outcomes. The purpose of this study was to assess university students’ mental health, specifically depression, anxiety, and stress, during Lebanon's extended COVID-19 pandemic, as well as the sociodemographic factors and lifestyle practices associated with it. 

Methods

An online anonymous survey assessed the rates of mental health problems during COVID-19, controlling for socio-demographics and other lifestyle practices, in 329 undergraduate and graduate university students. Instruments utilized were the Patient Health Questionnaire (PHQ-9) for depression, the Beck Anxiety Inventory (21-BAI) for anxiety, and the Perceived Stress Scale (PSS-10) for stress. The study employed descriptive statistics and multiple logistic regression models to analyze the association between depression, anxiety, and stress with sociodemographic and lifestyle factors. Results were evaluated using adjusted odds ratios and confidence intervals, with a significance level of 0.05.

Results

Moderate to severe rates of depression, anxiety and stress among students were reported by 75.9%, 72.2%, and 89.3%, respectively. The odds of anxiety and stress were higher among women compared to men. Students who used private counseling services had higher odds of anxiety and stress than those who did not. Overall rated health was a major predictor of depression and anxiety, with the "poor" and "fair" overall-reported health groups having higher odds than the "Excellent" group. When compared to those who did not smoke, students who increased their smoking intake had higher odds of depression, anxiety and stress. Students who reduced their alcohol consumption had lower odds of anxiety compared to those who did not consume alcohol. Students who reduced their physical activity had higher odds than those who increased it. Finally, students who slept fewer than seven hours daily had higher odds of depression than those who slept seven to nine hours.

Conclusion

Our findings indicate a national student mental health crisis, with exceptionally high rates of moderate to severe depression, anxiety, and stress. Factors such as gender, university program, overall rated health, importance of religion in daily decisions, private counseling, smoking cigarettes, alcohol consumption, physical activity, and sleeping, were all found to have an impact on mental health outcomes. Our study highlights the need for university administrators and mental health professionals to consider targeted mental health programming for students, particularly for women and those with poor or fair overall perceived health.

2. In the body of the manuscript, in Method, it is necessary for the authors to explain how, once data collection was carried out through a Forms questionnaire, the option for a larger sample would pose a risk of spreading the Covid-19 virus.

Authors’ response: 

Thank you for your input. We have revisited the Method section and made adjustments to address your concern.

The questionnaire was distributed in Lebanon in Arabic and English. Prior to filling the survey, all participants provided written informed consent online. To adapt to the rapidly changing pandemic context and prioritize participant and researcher safety, we employed an online convenience sampling approach. This method was chosen over traditional in-person methods to minimize potential transmission, particularly given the rapid spread associated with the Omicron variant. The decision to use online distribution platforms aligns with previous methodologies adopted in COVID-19-related research [28, 36, 37].

3. Still in Method, it is necessary for authors to justify the choice of the PHQ-9, BAI and PSS instruments and not others, with the same purpose. It is also necessary to provide information on whether the instruments used were validated in the language, in addition to English, which was used by the authors.

After these adjustments, the manuscript must be reevaluated.

Authors’ response: 

Thank you for your valuable feedback. We acknowledge the importance of justifying our choice of the PHQ-9, BAI, and PSS instruments, particularly with regard to their relevance and validity for our study population.

It is important to note that these instruments have been validated among university students, which aligns closely with the demographic of our study. Furthermore, we have ensured the linguistic validity of these instruments by utilizing validated Arabic translations. We will incorporate these justifications into the Method section of the manuscript, along with information on the validation of these instruments in Arabic, as suggested. We appreciate your guidance on this matter.

PHQ-9

“PHQ-9 is a self-rating scale with strong reliability and validity for students [41, 42].”

“The Arabic translated version of the PHQ-9, which has been validated, demonstrated good reliability with a Cronbach alpha of 0.88 [43]. In our study, the Cronbach's alpha coefficient of the PHQ-9 was 0.901.”

21-BAI

“The Arabic-translated version of the 21-BAI scale has been validated among university students in Kuwait, with Cronbach's alpha estimated to be between 0.83 and 0.90 [49]. In our study, the Cronbach's alpha coefficient of the BAI scale was 0.944.”

PSS-10

“The PSS is a simple global stress measure that has been proven to be reliable and valid in a variety of settings and languages [52-55]. In particular, the PSS-10 questionnaire was validated to assess stress among university students in a study conducted in China [56]. The Arabic version of the PSS-10 was validated in a study conducted in Lebanon, demonstrating good Cronbach's alpha reliability (0.74) [57]. The Cronbach's alpha coefficient of the PSS-10 scale was 0.846 in this study.”

Reviewer #2:

 The submitted paper is a cross-sectional study of depression, anxiety, stress, and life factors in students during the COVID-19 pandemic, and the paper contains many important findings that will be useful to readers.

1. I have no opinion on major revisions to the content of the text, but the inclusion of references in the text needs to be corrected. Specifically, the year of publication needs to be added to the references in the text, as sometimes only the author's name is given and not the year of publication. For example, the article by Maalouf et al. and the article by Rogowska et al.

Authors’ response: 

Thank you for your valuable feedback. We appreciate your attention to detail regarding the references in the text. We have addressed this issue by adding the year of publication to all references where applicable.

2. Overall, the resolution of the figures is poor and I do not think they are of publishable quality. The resolution of the figures needs to be improved.

Authors’ response: 

Thank you for your input. We have reviewed the figures, and while we believe they are of high quality, we understand your concern. It seems the resolution might not be coming through well in the PDF. We recommend downloading the figures for better clarity.

---

## [Decision Letter · Decision Letter 1]

20 Mar 2024

PONE-D-23-37343R1A Cross-sectional Study of University Students' Mental Health and Lifestyle practices Amidst the COVID-19 PandemicPLOS ONE

Dear Dr. Hoteit,

Thank you for submitting your manuscript to PLOS ONE. After careful consideration, we feel that it has merit but does not fully meet PLOS ONE’s publication criteria as it currently stands. Therefore, we invite you to submit a revised version of the manuscript that addresses the points raised during the review process.

We look forward to receiving your revised manuscript.

Kind regards,

Amin Nakhostin-Ansari

Academic Editor

PLOS ONE

Journal Requirements:

Reviewers' comments:

Reviewer's Responses to Questions

**Comments to the Author**

1. If the authors have adequately addressed your comments raised in a previous round of review and you feel that this manuscript is now acceptable for publication, you may indicate that here to bypass the “Comments to the Author” section, enter your conflict of interest statement in the “Confidential to Editor” section, and submit your "Accept" recommendation.

Reviewer #1: All comments have been addressed

Reviewer #2: All comments have been addressed

2. Is the manuscript technically sound, and do the data support the conclusions?

Reviewer #1: Partly

Reviewer #2: (No Response)

3. Has the statistical analysis been performed appropriately and rigorously? 

Reviewer #1: Yes

Reviewer #2: (No Response)

4. Have the authors made all data underlying the findings in their manuscript fully available?

Reviewer #1: Yes

Reviewer #2: (No Response)

5. Is the manuscript presented in an intelligible fashion and written in standard English?

Reviewer #1: Yes

Reviewer #2: (No Response)

6. Review Comments to the Author

Reviewer #1: It is suggested that authors delete the following sentence "Scalable solutions, such as online mindfulness (Ahmad et al., 2018; El Morr et al., 2017; ElMorr et al., 2020), are also important to investigate in order to alleviate the mental health crisis among university students in Lebanon" from the Conclusion, as the content is not properly addressed in the manuscript. Furthermore, references have no place in Conclusions

Reviewer #2: (No Response)

7. PLOS authors have the option to publish the peer review history of their article (what does this mean?). If published, this will include your full peer review and any attached files.

Reviewer #1: **Yes: **Rosana Cipolotti

Reviewer #2: No

---

## [Author Response · Author response to Decision Letter 1]

22 Mar 2024

22-March-2024

PLOS ONE

Re Manuscript ID: PONE-D-23-37343R1 entitled “A Cross-sectional Study of University Students' Mental Health and Lifestyle practices Amidst the COVID-19 Pandemic.”

We would like to thank the editor and reviewers for the opportunity to review our manuscript entitled: “A Cross-sectional Study of University Students' Mental Health and Lifestyle practices Amidst the COVID-19 Pandemic.”

We deeply appreciate all the helpful feedback and suggestions that helped us enhance the article's content. We responded to all of the journal’s and reviewers’ comments and made significant improvements.

Our responses (in bold black and blue, with blue indicating newly included or modified statements in the paper) are described below in a point-to-point manner.

Authors’ response: 

We have carefully reviewed our reference list and made the necessary adjustments as per your instructions. Specifically, we have replaced the reference to the article titled "Reliability and validity of the Patient Health Questionnaire-9 in Chinese adolescents" by Hu et al. with another relevant reference:

Original Reference: Hu X, Zhang Y, Liang W, Zhang H, Yang S. Reliability and validity of the patient health questionnaire-9 in Chinese adolescents. Sichuan Ment Health. 2014;27(4):357-60.

Replacement Reference: Reference #42: Zhang YL, Liang W, Chen ZM, Zhang HM, Zhang JH, Weng XQ, et al. Validity and reliability of Patient Health Questionnaire-9 and Patient Health Questionnaire-2 to screen for depression among college students in China. Asia-Pacific Psychiatry. 2013;5(4):268-75.

Reviewer #1: 

Reviewer #1: It is suggested that authors delete the following sentence "Scalable solutions, such as online mindfulness (Ahmad et al., 2018; El Morr et al., 2017; ElMorr et al., 2020), are also important to investigate in order to alleviate the mental health crisis among university students in Lebanon" from the Conclusion, as the content is not properly addressed in the manuscript. Furthermore, references have no place in Conclusions.

Authors’ response: 

Thank you. Please note that we have deleted the following sentence from the conclusion.

“Scalable solutions, such as online mindfulness [92-94], are also important to investigate in order to alleviate the mental health crisis among university students in Lebanon.”

Reviewer #2: (No Response)

Authors’ response: 

Thank you for taking the time to review our manuscript.

---

## [Decision Letter · Decision Letter 2]

1 Apr 2024

A Cross-sectional Study of University Students' Mental Health and Lifestyle practices Amidst the COVID-19 Pandemic

PONE-D-23-37343R2

Dear Dr. Hoteit,

We’re pleased to inform you that your manuscript has been judged scientifically suitable for publication and will be formally accepted for publication once it meets all outstanding technical requirements.

Kind regards,

Amin Nakhostin-Ansari

Academic Editor

PLOS ONE

Additional Editor Comments (optional):

Reviewers' comments:

Reviewer's Responses to Questions

**Comments to the Author**

1. If the authors have adequately addressed your comments raised in a previous round of review and you feel that this manuscript is now acceptable for publication, you may indicate that here to bypass the “Comments to the Author” section, enter your conflict of interest statement in the “Confidential to Editor” section, and submit your "Accept" recommendation.

Reviewer #1: All comments have been addressed

2. Is the manuscript technically sound, and do the data support the conclusions?

Reviewer #1: Yes

3. Has the statistical analysis been performed appropriately and rigorously? 

Reviewer #1: Yes

4. Have the authors made all data underlying the findings in their manuscript fully available?

Reviewer #1: Yes

5. Is the manuscript presented in an intelligible fashion and written in standard English?

Reviewer #1: Yes

6. Review Comments to the Author

Reviewer #1: All comments have been addressed by the authors. I do not have more comments.

All comments have been addressed.

7. PLOS authors have the option to publish the peer review history of their article (what does this mean?). If published, this will include your full peer review and any attached files.

Reviewer #1: **Yes: **Rosana Cipolotti

---

## [Editor Report · Acceptance letter]

3 Apr 2024

PONE-D-23-37343R2 

PLOS ONE

Dear Dr. Hoteit, 

I'm pleased to inform you that your manuscript has been deemed suitable for publication in PLOS ONE. Congratulations! Your manuscript is now being handed over to our production team.

Kind regards, 

on behalf of

Dr. Amin Nakhostin-Ansari 

Academic Editor

PLOS ONE